# Molecular Mechanisms and Potential Rationale of Immunotherapy in Peritoneal Metastasis of Advanced Gastric Cancer

**DOI:** 10.3390/biomedicines10061376

**Published:** 2022-06-10

**Authors:** Donghoon Kang, In-Ho Kim

**Affiliations:** 1Division of Gastroenterology, Department of Internal Medicine, Seoul St. Mary’s Hospital, The Catholic University of Korea, Seoul 06591, Korea; etiria@catholic.ac.kr; 2Division of Medical Oncology, Department of Internal Medicine, Seoul St. Mary’s Hospital, The Catholic University of Korea, Seoul 06591, Korea

**Keywords:** gastric cancer, peritoneal metastasis, immune checkpoint inhibitors

## Abstract

Peritoneal metastasis (PM) is one of the most frequent metastasis patterns of gastric cancer (GC), and the prognosis of patients with PM is very dismal. According to Paget’s theory, disseminated free cancer cells are seeded and survive in the abdominal cavity, adhere to the peritoneum, invade the subperitoneal tissue, and proliferate through angiogenesis. In these sequential processes, several key molecules are involved. From a therapeutic point of view, immunotherapy with chemotherapy combination has become the standard of care for advanced GC. Several clinical trials of newer immunotherapy agents are ongoing. Understanding of the molecular process of PM and the potential rationale of immunotherapy for PM treatment is necessary. Beyond understanding of the molecular aspect of PM, many studies have been conducted on the modality of treatment of PM. Notably, intraperitoneal approaches, including chemotherapy or immunotherapy, have been conducted, because systemic treatment of PM has limitations. In this study, we reviewed the molecular mechanisms and immunologic aspects of PM, and intraperitoneal approaches under investigation for treating PM.

## 1. Introduction

In 2020, gastric cancer (GC) was responsible for more than one million new cancer diagnoses and approximately 769,000 deaths. It is the fifth most common cancer and the fourth leading cause of death worldwide [1]. GC commonly metastasizes to the liver, peritoneum, lungs, bones, and lymph nodes. Of which, the peritoneum is one of the most frequent sites of metastasis in patients with GC. Peritoneal metastasis (PM) occurs synchronously with primary GC in about 14–43% of the patients. Meanwhile, PM occurs metachronously in about 10–46% of the patients who have undergone curative surgery [2,3]. Moreover, the prognosis of patients with PM is very dismal. According to Paget’s “seed and soil” theory [4], PM is initiated by the spread of tumor cells. These detached viable cancer cells, which are analogous to the “seed”, are in an appropriate microenvironment, and colonize a compatible organ, which is analogous to the “soil”. Several studies have been conducted to elucidate this multistep process. Cancer cells detach from the primary tumor, survive in the microenvironment of the abdominal cavity, attach to peritoneal mesothelial cells, invade the basement membrane, settle, and proliferate with angiogenesis. Each step of PM progression in gastric cancer depends on several molecular mechanisms, but the molecular mechanisms of these steps remain poorly understood.

The success of immunotherapy, including immune checkpoint inhibitors (ICIs), has changed the treatment landscape of several cancers, and ICIs have shown promising results in treating GC. Therefore, there is a need to understand the immunologic microenvironment of PM, and to develop a promising treatment strategy for PM.

In this study, we focus on the molecular mechanisms and immunological aspects of PM in GC, and intraperitoneal approaches under investigation for treating PM.

## 2. Multistep Process

### 2.1. Detachment of Cancer Cells and Transmigration to the Peritoneum

The first step in peritoneal dissemination is the detachment of cancer cells from the primary gastric tumor. Tumor cells that are detached from the primary tumor mass invade through the gastric wall to gain access to the peritoneal cavity. Cancer cells must be able to migrate and infiltrate to detach successfully from the original tumor (Table 1).

Many cytokines, growth factors, chemokines, and proteases are abundant in primary tumors, promote tumor cell survival and proliferation, and allow the tumor cells to migrate. Epithelial–mesenchymal transition (EMT) is a process in which a subset of tumor cells in the primary tumor switches off epithelial markers such as E-cadherin, and turns on mesenchymal markers such as S100 calcium-binding protein A4 (S100A4) [5] and vimentin, resulting in cell polarity loss, cytoskeletal reorganization, and the dissolution of adherens and tight junctions [6,7]. E-cadherin is a calcium-dependent cell–cell adhesion molecule that is essential for epithelial architecture, cell polarity, and differentiation maintenance. When dysregulated, cell motility is promoted, leading to tumor invasion and peritoneal dissemination [8]. In addition, E-cadherin and the cadherin–catenin complex accelerate cell motility and invasiveness by modulating various signaling pathways in epithelial cells, such as Wnt, Rho GTPase, and NF-κB signaling pathways, as well as EMT [9,10,11,12] (Figure 1).

Matrix metalloproteinases (MMPs) are proteolytic enzymes that play a role in the degradation of extracellular matrix (ECM) proteins. The overexpression of MMPs in cancer cells causes uncontrolled proteolytic activity, tissue remodeling, and excessive basement membrane destruction, allowing tumor cells to gain stromal access [13]. High expression of MMP-2 and MMP-9 is associated with invasiveness and poor survival in patients with GC [14]. MMP-7 is also highly expressed in GC cells [15,16] and is associated with cancer aggressiveness [17] and peritoneal dissemination [18].

EMT is a crucial step in the initiation of local invasion and subsequent dissemination. During EMT, tightly kinked epithelial cells separate into motile and invasive mesenchymal cells. At the mesenchymal stage, cancer cells have a new ability to penetrate the surrounding milieu. The ligands EGF, TGFβ, Wnt, Notch, and integrin play a significant role in EMT [10].

### 2.2. Survival in the Peritoneal Cavity Microenvironment

Peritoneal dissemination is initially driven by direct invasion of tumor cells from the gastric wall to the peritoneal cavity, including spontaneous spreading from the primary tumor, or surgical trauma causing scattering of cancer cells during surgery. These detached cells are called intraperitoneal-free cancer cells (IFCC), and encounter a hypoxic and glucose-deficient microenvironment in the peritoneal cavity. They must have the ability to survive, migrate, and proliferate within this milieu. 

Hypoxia-inducible factor-1α (HIF-1α) is a crucial transcription factor involved in angiogenesis and glycolysis in the cellular response to hypoxia. HIF-1α stimulates the expression of various genes involved in the adaptation to hypoxia and glucose metabolism. HIF-1α promotes EMT in cancer cells by activating the transcription of genes in the LOX family [27,28,29], and induces angiopoietin-like-4 (ANGPTL4), resulting in tumor growth and resistance to anoikis [45,46,47].

Anoikis, programmed apoptosis triggered by cell detachment from the extracellular matrix (ECM)—is another important barrier to the survival of detached cells and their reattachment to the new matrix in ectopic sites [65]. Cancer cells must develop resistance to anoikis for tumor progression. Anoikis resistance (AR) is a prerequisite for hematogenous metastasis [65,66], lymphatic metastasis [7], and PM of GC [67]. Several processes contribute to AR development, including promotion of EMT, oncogene activation, adaptation of metabolism, and changes in expression of the integrin family of genes [49,68]. The PI3K/Akt, PTEN/PI3K/NF-kB/FAK, and CXCL12/CXCR4 pathways are associated with anoikis resistance [30,31,32,33,34,35,36,37,38].

### 2.3. Attachment of Free Tumor Cells to Peritoneal Mesothelial Cells or Lymphatic Stomata

In PM formation, the attachment of GC cells to the peritoneal lining is crucial. There are two distinct pathways for peritoneal dissemination: transmesothelial and translymphatic, which is depicted in Section 2.4 [69,70,71,72].

In transmesothelial metastasis, cancer cells seeded in the peritoneal cavity adhere to the peritoneal surface directly [73]. To penetrate the submesothelial area, the mesothelium must overcome a barrier. The peritoneum is a lining of a single layer of closely connected mesothelial cells, which creates an anatomical barrier for the prevention of cancer cell invasion [74]. Most IFCCs attached to mesothelial cells die due to this mechanical barrier and a hostile microenvironment that lacks nutrients. 

Some cancer cells evade this process by producing growth factors and matrix metalloproteinases. The growth factor released by IFCCs causes peritoneal mesothelial cells to undergo EMT and transform into a spindle-like, fibroblastic-pattern morphology, increasing the space between the mesothelial cells and exposing the basement membrane [75,76]. In the process of EMT, transforming growth factor β1 (TGF- β1) and integrins play a crucial role.

TGF-β1 has been reported to control the proliferation and differentiation of cells. High TGF-β1 levels in the peritoneal washing fluid have been reported to be associated with PM formation. Furthermore, EMT was induced by TGF-β1, which provides a suitable condition for “soil”. Several factors that regulate TGF-β1 activity have been identified and are being investigated as potential anticancer molecules for the prevention of PM in GC. For example, bone morphogenic protein and activin membrane-bound inhibitor (BAMBI) inhibited the TGF-β/EMT signaling pathway and suppressed the invasiveness of gastric tumors [77]; ASPP2 inhibited TGF-β1-induced EMT in GC cells by inhibiting in them the phosphorylation and nuclear accumulation of Smad2/3 [78]; and Ki26894, a TβR-I kinase inhibitor showed a decrease in invasiveness and EMT in GC [79]. These results imply the possibility of promising drugs targeting TGF-β1.

Furthermore, the exposure of the basement membrane of peritoneal mesothelial cells is mediated by integrin molecules. Integrins are membrane-bound proteins that directly contact cells and the ECM, and serve as adhesion receptors for ECM proteins and cellular counter ligands. In GC, α1, α2, α3, and β1 subunits have been reported to be closely associated with PM formation; α3β1 integrin (VLA-3) is especially associated with GC adhesion to laminin 5, a major ECM glycoprotein. It has been reported that the connective tissue growth factor (CTGF) effectively blocks adhesion by binding to VLA-3, suggesting a potential therapeutic role for recombinant CTGF [61].

### 2.4. Invasion into Subperitoneal Space

After successful attachment, IFCCs degrade the ECM and the peritoneal blood barrier, and invade deeper into the subperitoneal tissue, at which point, connective tissue underneath the mesothelium helps to create a niche for the seeding of cancer nodules. MMPs and integrins are crucial in PM. Once IFCCs loosely attach to mesothelial cells with adhesion molecules such as CD44, cytokines are released, to contract mesothelial cells by phosphorylation of their cell skeleton [72].

Cancer cells can synthesize MMPs and degrade ECM. By degrading ECM proteins and regulating the activity of other biomolecules, MMP7 is regarded as a central molecule associated with the stromal invasion of GC cells and PM formation [80,81,82]. 

Lymphatic orifices on the peritoneal surface open into the peritoneum. Translymphatic metastasis can occur when IFCCs move into lymphatic orifices and nourish the submesothelial lymphatic space beneath the lymphatic stomata [83,84]. Because of the lack of a physical barrier in the peritoneal mesothelial cell layer, translymphatic metastasis frequently develops at an earlier stage than transmesothelial metastasis. A gate, through which small particles are absorbed from the peritoneal cavity into the subperitoneum, is considered the so-called milky spot, which is the lymphoid tissue on the peritoneum. These lymphatic orifices are located in the greater omentum, the inferior surface of the diaphragm, the small bowel mesentery, the pelvic peritoneum, and the falciform ligament, whereas the anterior abdominal wall, the liver capsule, and the serosal surface of the stomach and small bowel are rarer locations of lymphatic orifices. Consequently, locations with many lymphatic orifices are invaded in the early stages of peritoneal metastasis; however, areas with few orifices are unaffected until the later stages of peritoneal metastasis [85,86].

Milky spots are aggregations of macrophages and lymphocytes that help remove particles, germ cells, and tumor cells from the peritoneal cavity, and play an essential role in peritoneal defense. However, in the early phases of peritoneal dissemination, cancer cells preferentially infiltrate into milky areas and seek a milieu in which they may live, develop, and form solid metastases [58,59,60,61]. The presence of unique features in milky spots, such as increased amounts of cellular adhesion molecules and growth-stimulatory proteins, might explain this irony [61,62]. Within milky patches, detached GC cells (seeds) find a microenvironment with favorable physical and chemical characteristics that allows them to survive and proliferate, forming cluster-type metastases. Tumor-associated macrophage (TAM)-induced peritoneal mesothelial cell fibrosis [87], CCL22/CCXR4 axis [88], and HIF-1α [89] have been reported to facilitate tumor cell invasion.

### 2.5. Proliferation with Blood Vascular-Neogenesis

Angiogenesis is a key step in the various stages of human cancer development and dissemination. When IFCCs invade near the subperitoneal capillaries, they induce angiogenesis by proliferation via autocrine or paracrine loops through the production of growth factors. Vascular endothelial growth factor (VEGF), which is one of the most potent angiogenic molecules secreted from cancer cells, enhances tumor growth by inducing neoangiogenesis in the peritoneal microenvironment [63], and promotes vascular permeability in the peritoneum. Previous research has suggested that VEGF is linked to PM in GC. Antisense therapy of the VEGF receptor has been shown to diminish angiogenesis and PM in GC [62,90,91]. As a result, the integrity of the peritoneal blood barrier is disrupted, to create a ready soil for establishing PM. Treatments targeting VEGF are being attempted, and drugs such as ramucirumab, a monoclonal antibody that binds to VEGF-R2 and prevents its activation, are being tried to treat advanced GC [90].

## 3. Potential Rationale of Immunotherapy in PM of GC

Recently, immunotherapy, including programmed death/programmed death-ligand 1 (PD-1/PD-L1) inhibitors, has led to many advances in cancer treatment. ICI treatment has also shown promising results in advanced GC, and several newer immunologic agents are under investigation. Therefore, it is important to understand the immunological characteristics of PM, so as to usher in the future of immuno-oncology. 

Immune cells can be involved in innate and adaptive immune systems, and exhibit antitumor activity. However, there are few studies on the role of immune cells in the PM in GC. As it is known that immune cells such as macrophages and lymphocytes are present in greater omentum and lymph nodes [91], then immune cells are potential candidates for PM treatment.

Macrophages not only play a role in chronic inflammation but also initiate, promote, or suppress the development of cancer by phagocytosis, antigen presentation, and production of cytokines and growth factors that affect other immune cells [92]. In the early stages of cancer, tumor-associated macrophages (TAMs) appear to have an inflammatory, tumoricidal phenotype called M1-macrophages. M1-macrophages have phagocytic and antigen-presenting activities, produce pro-inflammatory cytokines, exert cytotoxic effects on tumor cells, and promote indirect cytotoxicity by activating other immune cells, such as natural killer (NK) cells and T-cells [93]. In contrast to M1, during tumor progression, most macrophages switch to the M2 phenotype followed by their interactions with tumor cells. M2 phenotype has a repertoire of tumor-promoting capabilities involving immunosuppression, angiogenesis, and neovascularization, as well as stromal activation and remodeling [94] (Figure 2). A previous study characterized TAM using single-cell RNA sequencing in the malignant ascites of GC. The study suggested that TAMs from malignant ascites in GC have strong M2-like characteristics, and that this M2-like phenotype of TAMs is associated with poor prognosis [95]. The study also showed that macrophages from ascites of GC showed the most M2-like features compared to macrophages from other cancer types. Another study investigated the role of TAMs in patients with PM of GC [96]. The study investigated TAMs using immunohistochemistry in the primary tumor, surgical margin, PM lesions, and adjacent peritoneal tissue. In the study, patients with PM showed an increased number of TAM and M2 macrophages, upregulated levels of angiogenesis in the peritoneum and macrophages, and increased levels of epidermal growth factor and vascular endothelial growth factor-expressing macrophages. Although the mechanical aspect of TAM as a potential therapeutic target is still not well understood, previous studies have shown that M2-like macrophage infiltration is highly associated with PD-L1 expression in GC cells, and that extracellular vesicles derived from GC play a role by affecting macrophage phenotypes [97,98]. Considering these results, immunotherapy or cancer vaccines targeting TAM may be a promising strategy for PM treatment. 

NK cells are important effectors of anticancer immune response, and can survey and control tumor initiation because of their ability to recognize and kill malignant cells, and regulate the adaptive immune response via cytokine and chemokine release [99]. Studies have shown that NK cells are activated by several cytokines, immunomodulatory drugs, immune checkpoint blockades, antibodies, vaccines, and gene therapy in GC [100]. One previous study evaluated gene therapy with an adenovirus vector that expresses high levels of intercellular adhesion molecule-2 (ICAM-2) in the human GC cell line OCUM-2MD3, which has high peritoneal metastatic ability in nude mice [101]. ICAM-2, a ligand of CD11a/CD18 (LFA-1), is mainly expressed on endothelial and hematopoietic cells, and can activate and migrate NK cells [102]. The study showed that the transduction of ICAM-2 into cancer cells enhances the adhesion and activation of NK cells, resulting in reduced PM. Another study investigated the role of cancerous immunoglobulin (Ig) in cancer cell growth in GC cells transfected with cancerous IgG heavy chain small interfering RNA (siRNA). Cancerous Ig reduced antibody-dependent cell-mediated cytotoxicity (ADCC) induced by an anti-human epithelial growth factor receptor (EGFR) antibody, suggesting that the cancerous Ig-Fc receptor interaction inhibits the NK cell effector function [103]. A recent study evaluated chimeric antigen receptor (CAR)-NK cells targeting mesothelin, a cell-surface glycoprotein with normal expression restricted to mesothelial cells lining the peritoneum [104]. This study constructed mesothelin and CD19-targeted CAR-NK cells, and demonstrated that mesothelin-CAR NK cells could effectively eliminate GC cells in both subcutaneous and intraperitoneal tumor models. 

CAR-based strategies are now being studied more in the form of CAR-T-cell treatment. CAR-T-cell treatment is an emerging strategy, and CARs are engineered synthetic receptors that redirect lymphocytes, most commonly T-cells, to recognize and eliminate cells expressing a specific target antigen [105]. CAR-T-cell treatment designed for several different targets—including epithelial cell adhesion molecule (EpCAM), human epidermal growth factor receptor 2 (HER2), mesothelin, and carcinoembryonic antigen (CEA) in GC—has proven useful in experimental studies, and is currently being evaluated in registered clinical trials including patients with advanced GC [106]. Recently, a phase 1 trial investigated anti-claudin (CLDN)18.2 CAR-T-cell therapy in gastrointestinal cancer, including GC [107]. In heavily treated patients with GC, anti-CLDN18.2 CAR-T-cell therapy showed promising efficacy (objective response rate [ORR], 61.1%; median progression-free survival/overall survival (PFS/OS), 5.4/9.5 months). Most studies have been conducted with IV administration of CAR-T therapy, and the intraperitoneal approach may be a potential strategy for CAR-T. The first intraperitoneal delivery of CAR-T-cells was reported by Katz, et al., [108]. This study investigated intraperitoneal anti-CEA CAR-T-cells in mice with colorectal cancer and PM. In this study, intraperitoneal anti-CEA CAR-T-cells resulted in superior tumor reduction and a durable response compared with systemic infusions. Several studies on intraperitoneal CAR-T-cells have shown promising results in murine models [109,110]. Therefore, the feasibility of CAR-T-cells in the treatment of PM should be confirmed by the results of several ongoing clinical trials. 

In addition to using effector T-cells such as CAR-T, strategies for inhibiting cells—such as tumor infiltrative regulatory T-cells (Treg) and several immune checkpoints, which play immunosuppressive roles—are also being studied. Tregs are important factors in the immune microenvironment of GC. (Figure 3) In addition, immunosuppressive lymphocytes exert negative immunoregulatory effects by regulating the active immune function of effector T cells [111]. A previous study investigated T cell subsets in lymphocytes derived from malignant ascites, and the effects of arsenic trioxide (As_2_O_3_) on Tregs and ascites-derived tumor-infiltrating lymphocytes (TILs) in vitro [112]. This study suggests that As_2_O_3_ may induce selective depletion and inhibit the immunosuppressive function of Tregs, and may enhance the cytotoxic activity of ascites-derived TILs.

Several immune checkpoint molecules are being studied in GC. Several clinical trials of ICIs including PD-1/PD-L1 inhibitors have shown promising results [113,114,115,116,117,118] in advanced GC. ATTRACTION-2, a randomized, double-blind, placebo-controlled, phase 3 trial, assessed the efficacy and safety of nivolumab (anti-PD-1 inhibitor) vs. placebo in heavily pretreated patients with advanced GC [118]. This study demonstrated that nivolumab showed OS benefit compared to placebo. The 12-month OS rates were 26.2% with nivolumab, and 10.9% with the placebo. Recently, in the first-line setting, ICIs combined with chemotherapy have shown clinical efficacy. CheckMate 649 was a randomized, open-label, phase 3 trial that compared nivolumab plus chemotherapy, nivolumab plus ipilimumab (anti-cytotoxic T-lymphocyte-associated antigen 4 [CTLA4] inhibitor), or chemotherapy alone in previously untreated, unresectable, HER2 negative gastric, gastroesophageal junctions, or esophageal adenocarcinoma [114]. This study demonstrated that nivolumab plus chemotherapy resulted in significant improvements in OS and PFS, compared with chemotherapy alone, in patients with a PD-L1 CPS ≥ 5, CPS ≥ 1, and all randomly assigned patients. ATTRACTION-4 was a randomized, multicenter, double-blind, placebo-controlled, phase 2–3 trial that investigated nivolumab with chemotherapy versus placebo with chemotherapy as first-line therapy for patients with HER2-negative, unresectable advanced or recurrent gastric or gastroesophageal junction cancer [115]. Although this trial did not meet the primary endpoint (OS), nivolumab combined with chemotherapy significantly improved PFS. This result is thought to be due to the absence of biomarkers and subsequent treatment, including immunotherapy, in the placebo group, because this trial was performed only in Asian patients. Currently, another large-scale phase III KEYNOTE-859 study which investigated pembrolizumab in combination with chemotherapy, as first-line treatment for patients with HER2 negative advanced unresectable or metastatic gastric/gastroesophageal junction adenocarcinoma, is ongoing [119]. The planned sample size was 1542 patients, and the primary endpoint was OS. Although these ICIs showed clinical benefit in advanced GC, it is not clear whether the clinical benefit of ICI is maintained in treating PM. To date, no study has evaluated the efficiency of ICI in patients with PM. However, this can be inferred from the subgroup analysis of large-scale clinical studies. An exploratory, post hoc subgroup analysis of ATTRACTION-2 showed that nivolumab did not show a clinical OS benefit in patients with PM (hazard ratio [HR] 0.74, 95% confidence interval [CI] 0.48–1.15) [118]. In addition, subgroup analysis of ATTRACTION-4 showed that nivolumab plus chemotherapy did not show a clinical benefit in terms of PFS and OS in patients with PM (PFS, HR, 1.04, 95% CI, 0.76–1.44; OS, HR, 1.20, 95% CI, 0.94–1.53) [115]. To date, few studies have evaluated the expression of immune checkpoint molecules such as PD-1/PD-L1 in the PM of GC. A previous study performed comprehensive immune profiling of PM specimens using a curated immune gene panel including markers for antigen presentation, B/T/macrophage/NK/MDSC lineages, co-stimulatory and co-inhibitory immune checkpoints and receptors, cytolytic activity, and activating cytokines [120]. This study separated PM specimens into two main groups, the T cell ‘exclusive’ and T cell ‘exhausted’ subtypes. The T cell ‘exhausted’ subtype showed high levels of immune checkpoint T-cell immunoglobulin and mucin-domain containing-3 (TIM-3), its ligand galectin-9, V-domain immunoglobulin suppressor of T-cell activation (VISTA), and transforming growth factor-β (TGF-β1); other classical checkpoints including PD-1, PD-L1/L2, CTLA-4, lymphocyte-activation gene 3 (LAG-3), indoleamine 2,3-dioxygenase 1 (IDO1), and T-cell immunoreceptor with Ig and ITIM domains (TIGIT), were low. Therefore, further studies should be performed to investigate other potential therapeutic immune checkpoints, as well as PD-1/PD-L1 as biomarkers for immunotherapy in PM. 

## 4. Intraperitoneal Approach to Treat GC with PM

Although systemic treatment is the standard option in metastatic [114,121] and adjuvant settings [122,123], the clinical benefit of systemic treatment is limited in patients with PM. Systemic treatment has several limitations in treating PM because of the presence of a plasma-peritoneal barrier that limits the access of intravenous chemotherapy to PM, and inadequate blood supply and oxygenation of the tumor cells, coupled with their low apoptotic potential [124]. A previous pharmacokinetic study suggested that intraperitoneal administration of a drug can maintain a significantly greater concentration in the peritoneal space than in plasma [125]. Several early-phase trials evaluated intraperitoneal chemotherapy in patients of GC with PM, and showed favorable results [126,127]. The results of a large-scale phase 3 trial, PHEONIX-GC, were reported in 2018 [128]. The patients were randomly assigned to receive intraperitoneal and intravenous paclitaxel plus S-1 or S-1 plus cisplatin. Unfortunately, this trial failed to demonstrate the statistical superiority of intraperitoneal paclitaxel plus systemic chemotherapy.

Intraperitoneal chemotherapy with hyperthermia enhances the penetration of chemotherapy into tumor tissues, and shows synergism with various chemotherapeutic agents [129]. To date, hyperthermic intraperitoneal chemotherapy (HIPEC) has been extensively studied, and the penetration depth of chemotherapy into tissue is limited; therefore, HIPEC is usually conducted with cytoreductive surgery (CRS) of all visible PM. Until now, the role of HIPEC has been controversial. Previous small-sized studies reported that HIPEC with CRS showed prolonged overall survival in patients with GC and PM [130,131,132]. A previous randomized phase 3 study suggested that CRS with HIPEC with mitomycin C and cisplatin may improve survival compared to CRS alone [133]. And a CYTO-CHIP propensity score matching study suggested that CRS-HIPEC improved OS and recurrence-free survival compared with CRS alone [134]. However, other studies have not shown the clinical benefits of HIPEC in GC with PM. Desiderio, et al., conducted a meta-analysis of 11 randomized controlled trials and 21 non-randomized control trials (2520 patients) of HIPEC for the treatment of GC [135]. This study defined the control group as patients who underwent CRS or systemic chemotherapy. This study showed no difference in the 2- and 3-year OS rates between HIPEC and control groups. Moreover, when comparing HIPEC with systemic chemotherapy, this analysis did not show a statistically significant difference between the groups. However, the HIPEC group showed a significantly higher risk of developing postoperative complications (relative risk [RR] = 2.15, 95% confidence interval [CI] 1.29–3.58) than did the control group. HIPEC was associated with a high risk of respiratory failure (relative risk [RR] = 3.67, 95% CI 2.02–6.67) and renal dysfunction (RR = 4.46, 95% CI 1.42–13.99, *p* < 0.01). Recently, Lei, et al., reported the results of HIPEC with systemic treatment [136]. This study used propensity matching analysis to compare HIPEC with systemic chemotherapy and systemic chemotherapy alone. This study suggests that HIPEC with chemotherapy has a significant survival benefit compared with systemic chemotherapy alone, for patients with GC and PM, without compromising patient safety. These conflicting results are thought to be due to the heterogeneous design of the HIPEC studies, the patient groups, and the HIPEC methods. Large-scale trials with good design are required in the future to establish HIPEC efficacy in treated patients with GC and PM.

Recently, systemic immunotherapy has shown promising results in GC; therefore, intraperitoneal immunotherapy can be a potential treatment strategy, and several approaches are under investigation. Catumaxomab, a bispecific (anti-EpCAM × anti-CD3) trifunctional antibody, was approved by the European Union in April 2009 for intraperitoneal treatment of patients with malignant ascites [137]. EpCAM is expressed in tumor cells but not in normal cells found in the peritoneal cavity lining and fluids (mesothelial cells, leukocytes, and macrophages) [138,139]. Therefore, EpCAM is a potential target for intraperitoneal antibody therapy. Catumaxomab combines the characteristics of classical monoclonal antibodies and bispecific molecules. Catumaxomab induces a tri-cell complex of tumor cells, T-cells, and accessory immune cells, due to the unique Fc composition of mouse IgG2a and rat IgG2b. Thereby, crosstalk between different types of redirected immune effector cells is initialized, which results in the efficient killing of tumor cells [140]. Therefore, intraperitoneal administration of catumaxomab offers the advantage of targeted, locoregional immunotherapy against EpCAM-positive tumor cells in the peritoneal cavity [141]. A previous randomized phase II trial investigated the efficacy of intraperitoneal catumaxomab followed by FLOT chemotherapy (5-fluorouracil, leucovorin, oxaliplatin, and docetaxel) and FLOT alone, in patients with GC and PM. The primary endpoint was the macroscopic complete remission (mCR) rate of PM [142]. This study showed a trend toward the superiority of intraperitoneal catumaxomab, but the difference of mCR rates was not statistically significant (mCR rates, 27% vs. 19%, *p* = 0.69). Currently, a randomized phase 3 trial comparing intraperitoneal catumaxomab and physician’s choice of treatment, in patients with GC with PM, is ongoing (NCT04222114). Another trial investigating intraperitoneal EpCAM CAR-T-cell treatment in patients with GC and PC is also ongoing (NCT03563326). This trial compared intraperitoneal EpCAM CAR-T-cell treatment to the physician’s choice of systemic chemotherapy. Until now, the intraperitoneal approach to immunotherapy has been scarce. Further studies should be conducted in the future.

## 5. Conclusions

Although there have been advances in systemic treatment in advanced/metastatic GC, the prognosis of patients with PM is very dismal. Understanding of molecular characterization in PM formation should be elucidated through future studies. Furthermore, it is necessary to understand the immunological characteristics of PM in GC, and further studies in the field of immune-oncology should be conducted. Finally, trials of the intraperitoneal approach, as well as systemic therapy for treating PM, are currently ongoing. We hope that these attempts will improve the prognosis of GC with PM.

## Figures and Tables

**Figure 1 biomedicines-10-01376-f001:**
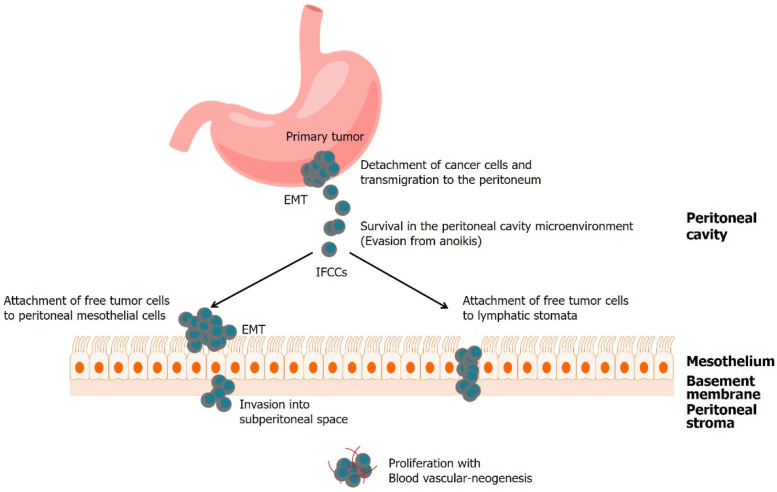
Process of peritoneal metastasis of gastric cancer.

**Figure 2 biomedicines-10-01376-f002:**
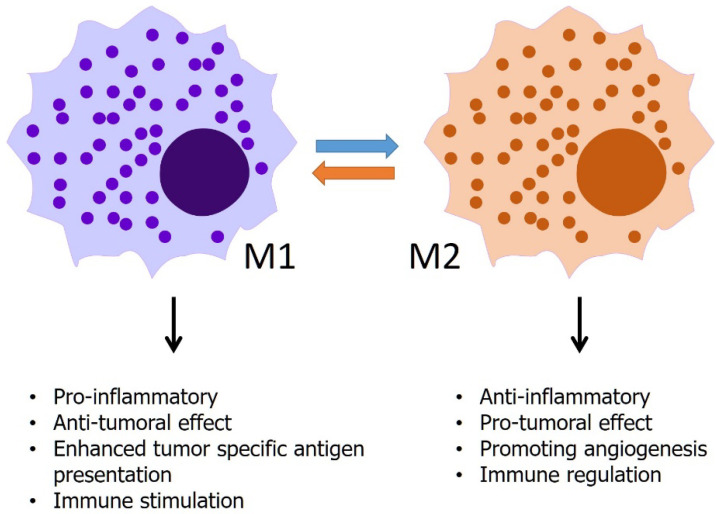
Differential role of M1 and M2 macrophages.

**Figure 3 biomedicines-10-01376-f003:**
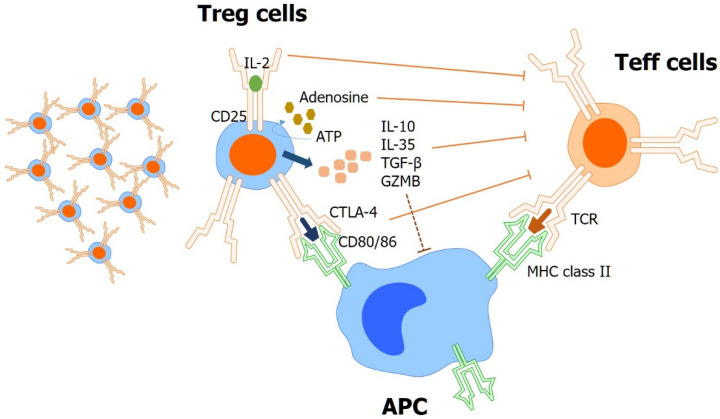
Role of Tregs in immune-evasion of cancer in tumor microenvironment. Regulatory T-cells (Treg) inhibit anti-tumor immunity by suppressing effector T-cells (Teff) in various mechanisms. (IL-2 depletion from surroundings by high-affinity binding of IL-2(CD25); abundant adenosine production by nucleotidase activity of CD39 and CD73; secreting IL-10, IL-35, and TGF-β; and transmission of suppressive signals by CTLA-4-CD80/86 binding of antigen-presenting cells (APC)).

**Table 1 biomedicines-10-01376-t001:** Molecules associated with PM in GC.

Molecule		Biological Function	Oncologic Function	Associated Molecules/Pathways	Ref.
Detachment of Cancer Cells and Transmigration to the Peritoneum					
CDH1	Cadherin 1, E-cadherin	Cell-cell adhesion	Proliferation, invasion, migration	Wnt, Rho GTPases, NF-kB pathways, EMT	[19,20]
ANXA1	Annexin 1	Calcium and membrane-binding protein	Proliferation, apoptosis, tumorigenesis	MAPK/ERK pathway	[21,22]
NRAGE	Neurotrophin receptor-interacting melanoma antigen-encoding gene homolog	Normal developmental apoptosis of sympathetic, sensory and motor neurons	Proliferation, apoptosis	AATF, p75NTR, PCNA	[23,24]
ARL4C	ADP-ribosylation factor-like 4C	GTP-binding protein	Promote cell motility	Rho GTPase, EGF, Wnt	[25,26]
Survival in the peritoneal cavity microenvironment					
HIF1A	Hypoxia-inducible factor 1alpha	Regulation of cellular and systemic homeostatic responses to hypoxia	Energy metabolism, angiogenesis, apoptosis	EMT, NF-kB pathway, glucose metabolism	[27,28,29]
PTEN	Phosphatase and tensin homolog	Dephosphorylating phosphoinositide substates	Growth, migration	PI3K/NF-kB pathway, FAK	[30,31,32]
Akt	Serine/threonine kinase	Receptor for pro-proliferation and bioactive substances, ECM receptor	Suppression of apoptosis, proliferation, metastasis, angiogenesis	PI3K/Akt, PTEN/PI3K/NF-kB/FAK	[33,34,35]
CXCR4/CXCL12	C-X-C motif chemokine receptor 4/Ligand12	Ligand, chemokine receptor	Invasion, metastasis, angiogenesis	EMT, CXCL12/CXCR4	[36,37,38]
AREG	Amphiregulin	Epidermal growth factor, mammary gland, oocyte and bone tissue development	Proliferation, migration	EGF, TGF-a, CXCL12/CXCR4 axis	[39,40,41]
LOX	Lysyl oxidase	Forming covalent crosslinks between collagen and elastic fibers	Invasion, metastasis	EMT	[42,43,44]
ANGPTL4	angiopoietin-like-4	Glucose metabolism	Induced by hypoxia, resistant to anoikis	FAK/Src/PI3K/Akt/ERK	[45,46,47]
MYH9	Myosin IIa or non-muscle myosin heavy chain 9 (NMMHC-IIA)	Cell motility, migration, adhesion	Resistance to anoikis	CTNNB1	[48]
C/EBPβ		Transcription factor	Induce PDGFB transcription	C/EBPβ-mediated-PDGFB autocrine and paracrine effects	[49,50]
Attachment of free tumor cells to peritoneal mesothelial cells or lymphatic stomata and invasion through the basement membrane					
TGF- β1	Tumor growth factor-beta1	Control proliferation and differentiation of cells	Normal development, wound healing	Smad	[51,52,53,54]
MMP7	Matrix metalloproteinase 7	ECM degradation	Proliferation, invasion	E-cadherin, TGF- β, EMT	[18,55,56]
CTGF	Connective tissue growth factor	Chondrocyte proliferation and differentiation, cell adhesion	Growth, migration, adhesion	Integrin α3β1, PDGF	[57,58]
MELK	Maternal embryonic leucine zipper kinase	Cell cycle-dependent protein kinase	Apoptosis, chemoresistance	RhoA, FAK, Bcl-GL	[23,59]
Integrin α3β1		Cell surface adhesion	Metastasis, adhesion	Lamine-5	[60,61]
Proliferation with blood vascular neogenesis					
VEGF	Vascular endothelial growth factor	Proliferation and migration of vascular endothelial cells	Angiogenesis	FAK, PI3K/AKT, MAPK/ERK	[62,63]
IRX1	Iroquois homeobox 1	Pattern formation in the embryo	Metastasis, angiogenesis	VEGFA	[23,64]

## Data Availability

Not applicable.

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
