# Peer review of "Molecular Mechanisms and Potential Rationale of Immunotherapy in Peritoneal Metastasis of Advanced Gastric Cancer"

_biomedicines, 2022, doi:10.3390/biomedicines10061376_

Round 1

Reviewer 1 Report

This is a well written comprehensive yet succinct review of the pathophysiology of peritoneal metastasis.  Obviously a whole textbook could be written on this but the authors did a nice job highlighting the main mechanisms and studies.  

The mechanisms of peritoneal mets are relevant to the second aim of this review and that is immunotherapy in peritoneal metastasis from gastric cancer.  The authors highlighted recent landmark studies such as Checkmate 649 and PHOENIX trial.  The review flowed nicely and was laid out in a logical format.  The authors included CAR-T cell, catumaxomab and IP chemotherapy. The review was balanced and fair.  

Author Response

Thank you for your kind comment.

Reviewer 2 Report

The topic of this work is relevant since peritoneal metastases form gastric cancer (GC) are one of the most difficult metastatic sites of this disease as well as a clinical and biological paradigm in other neoplasms. The Authors propose an easy-to-read but not superficial Review providing a panoramic view on the issue. In fact, the impact on prognosis and quality of life of this metastatic pattern of GC is enormous. The Review is simple, well organized and informative. The development of Authors' writing is consistent with the principal aim "the understanding of molecular process of PM and potential rationale of immunotherapy for PM treatment..." Epidemiology, molecular issues and clinical results are complete and well described.   Minor points:   1. I suggest an English language revision. In particular, I suggest not to start sentences with "And,...", this happens at lines 18, 42, 280.   2. Two additional figures at least depicting the differential role of M1 vs M2 macrophages and Tregs and the clash between PM and immune effector cells, should be added. This would increase the descriptive power and overall strength of this Review. 

Author Response

Thank you for your kind comments.

1) We revised the sentences that you pointed out.

2) We added two figures (Figure 2; different roles of M1 vs M2 macrophages, and Figure 3; role of Tregs in immune-evasion of cancer in tumor microenvironment) in order to increase descriptive power of the manuscript.